# Regional variation in knowledge and practice regarding common zoonoses among livestock farmers of selective districts in Nepal

**Kosh Bilash Bagale** [1]*, **Ramesh Adhikari**[2], **Devaraj Acharya**[3]

1 Graduate School of Education, Tribhuvan University, Kathmandu, Bagmati, Nepal, 2 Department of Geography and Population, Mahendra Ratna Campus, Tribhuvan University, Kathmandu, Bagmati, Nepal, 3 Bhairahawa Multiple Campus, Tribhuvan University, Bhairahawa, Rupandehi, Nepal

* koshbagale123@gmail.com

**Data Availability Statement:** All relevant data are included in the manuscript and it's supporting file.

**Funding:** The author(s) received no specific funding for this work.

## Abstract

### Background

The majority of Nepalese people are involved in farming. However, due to limited knowledge of zoonoses and poor preventive practices on the part of livestock farmers, vulnerabilities to zoonotic diseases are very high. The main objective of this study was to assess the regional variation in zoonoses-related knowledge and preventive practices of livestock farmers in different ecological regions of Nepal.

### Material and methods

Descriptive cross-sectional quantitative research design was followed in the study. The total sample size was 380 livestock farmers from randomly selected three ecological regions of Nepal. Systematic sampling techniques were applied for data collection. Data were entered into an excel sheet and then imported into Statistical Package for Social Sciences (SPSS) software. The data were calculated using descriptive statistics. Univariate, and bivariate analyses were performed, and the result of the study was presented in the form of text and tables based on their nature.

### Results

Of the studied six zoonotic diseases, most of the respondents (95.8%) knew about zoonotic bird flu; 90.7% of them, were about rabies; and 54.2% knew about swine flu. However, a few respondents knew about bovine tuberculosis, neurocysticercosis, and brucellosis. Ecologically, the highest number of respondents in Nawalpur had knowledge of rabies (95.3%), and swine flu (61.6%), whereas 98.3% of them had knowledge of avian influenza in Tanahun; and 12.5% of neurocysticercosis in Manang. Regarding zoonoses preventive practices such as regular hand washing with soap water, mask-wearing, gloves, boots, the respondents' representation of 60.8%, 6.6%, 1.8%, and 1.3% respectively in such practices show that although these are easy and cost-effective, personal protective equipment (PPE), such preventive practices were extremely underperformed. Not only that, only 12% of

**Competing interests:** The authors have declared that no competing interests exist.

respondents maintain a standard distance (>15m.) between their house and shed. Similarly, 17% still consumed meat from sick animals, and vaccination of livestock was also found poor coverage (36%) in the study.

## Conclusions

Livestock farmers need to be more knowledgeable about many common zoonotic diseases, and their preventive practices still need improvement, with significant regional variation in the study. This has invited various zoonosis threats for them. Therefore, it is recommended that the interventional programs related to common zoonoses be conducted for livestock farmers to solve the problem.

## Author summary

The majority of human diseases come from animal sources (zoonoses). People who are close to animals need to be knowledgeable about the nature of zoonoses and preventive measures. Among the common zoonoses in Nepal, only six diseases were delimited that were close to livestock. Due to neglected issues of zoonoses, we have limited data on the knowledge and preventive practices of livestock farmers, who are close to livestock for their livelihood. Therefore, this study seeks to reveal the realities of zoonoses-related knowledge and preventive practices among livestock farmers in selected districts of Nepal. Out of the six studied zoonoses, farmers have good knowledge of diseases that have a long endemic history in societies (i.e., rabies) or frequent outbreaks (i.e., avian influenza, swine flu). However, they are facing vulnerabilities due to poor preventive practices, and significant regional variations in their zoonoses-related knowledge and preventive practices. Lastly, this study might be significant for further planning in the public health, veterinary, and education sectors from the perspective of livestock farmers and indicate interventional programs against common zoonoses for vulnerable populations (i.e., livestock farmers).

## Introduction

Agriculture is the prime source of Nepal's economy, which equally contributes to the livelihoods of the farming communities, and has a significant impact on the National Gross Domestic Product [1]. According to an economic survey published by the ministry of finance, it is estimated that in the fiscal years 2020 /021, out of the total GDP, 20.2 percent of the contribution is covered by the agriculture sector [1]. In Nepal, approximately 66% of the population is still engaged in agriculture and livestock farming as an occupation [2], with the majority of these being mixed types of livestock farming, including poultry and buffalo. Ecologically, Nepal has divided into three regions, namely; Mountain, Hill, and Terai (plain region). The Mountain region is located in the northern part of the country and is attached to (Tibet) China, Hill covers the central part of the country and the plain region lies in the southern part of the country and is attached to India. Based on ecological and geographical diversity, livestock farmers in Nepal adopt different types of livestock in their households. Farmers in the Mountains keep Yak / Chauri and sheep whereas farmers in the Hills rear cows, sheep, goats,

and poultry, and in Terai, buffalo, cows, goats, and poultry are the major domesticated livestock in the country.

Due to globalization, industrialization and commercialization, paradigms have been shifting in the field of livestock farming as a commercial profession [3]. However, livestock farmers are still adopting traditional and vulnerable farming practices with low knowledge of livestock (vertebrate animals) related diseases called zoonoses that might be transmitted to human beings [4]. Furthermore, they use fertilizers, pesticides, or antibiotics in their agro-farming and livestock to get higher production, but they are ignoring their hazardous health implications during the use or handling of those things, including antibiotic resistance [5]. Without proper knowledge of zoonoses and their safety techniques, they are facing several vulnerabilities, including zoonotic diseases and huge human and economic losses.

Taylor and others mention that, more than 60% of infectious diseases in humans come from animal sources [6]. People who work with or near animals (livestock) are thus more vulnerable to zoonotic infections, which can be fatal. But people are ignoring such realities and also neglecting these issues in the country. In Nepal, many zoonotic diseases are prevalent in different forms, i.e., sporadic, endemic, epidemic, and so on [7]. When they get a favorable environment, that might change to a pandemic nature [3]. Therefore, to prevent zoonoses, we need to make knowledgeable at least those people who are close to and caretakers of their livestock (i.e., livestock farmers) through several interventional approaches (i.e., education or training) [8]. However, due to limited studies in the field of livestock farming, we have no actual ideas about the knowledge, and practices of livestock farmers related to zoonotic diseases and preventive practices. This study aims therefore to identify the realities of knowledge and existing preventive practices of livestock farmers in different ecological regions of Gandaki province of Nepal.

## Materials and methods

### Ethics statement

The study proposal was approved by the Research Committee Board of the Graduate School of Education, Tribhuvan University, Nepal [Ref: 68-076/77; 2076/12/5]. During the data collection, the ethical standard was maintained as per Nepal Health Research Council (NHRC) guidelines [9]. Verbal consent was taken before the interview and respondents were requested to participate voluntarily. All of the respondents were older than 18 years. All data were kept confidential with anonymity. Moreover, we also followed the ethical guidelines made by the American Psychological Association (APA) [10] throughout the research process.

This study follows the descriptive, cross-sectional quantitative research design. Face to face survey interview technique was employed for data collection. Respondents were selected by systematic random sampling technique from randomly selected three districts from three ecological regions namely; Manang, Tanahun, and Nawalpur districts of Gandaki province of Nepal.

### Population and sample size

The unit of analysis in the study was the household (HH). Therefore, the head of the household (HHH) or family members who have actively involved in livestock caring roles (including poultry and buffalo) were the populations that were surveyed. Based on the national population and housing census 2011, the total number of households (livestock farmers), included in all study districts (municipals), was 2835 [11]. The sample size in the study was calculated using the probability proportional to size (PPS) sampling technique proposed by Solvin [12], and the total sample size, including 10% for the non-response rate was 390. The following

formula suggested by Solvin, was used to calculate the sample size:

$$n = \frac{N}{1 + N(e2)} \qquad (1)$$

### Inclusion criteria

This study tried to assess the zoonoses related knowledge and preventive practices of live-stock farmers based on different ecological regions in Nepal. While exploring the real situation of livestock farmers towards zoonoses, researchers followed some inclusion criteria in the study. Among many zoonoses, this study only covers rabies, avian influenza, swine flu, brucellosis, bovine tuberculosis, and neuro-cysticercosis which are common zoonotic diseases related to livestock and poultry in Nepal [13]. Since there is only one district in the Terai region in Gandaki province, we picked for the reliability of data one district each from the hilly and mountain regions of the same province. Likewise, livestock (including buffalo and poultry) farmers who were the head or senior members of households, and actively involved in livestock farming were selected as the respondents on one respondent one household basis.

### Data collection tools, techniques, and analysis procedures

After taking the approval letter from the Graduate School of Education, Tribhuvan, University (GSE-TU), we visited the respective municipalities and took the official approval for data collection. Researchers spent almost 74 days in all study districts for data collection, and a total of 380 respondents participated in the survey interview with face-to-face techniques in their households. In the preliminary phase of the data collection procedure, we established a good rapport with the participants and then explained the research objectives. Standardized tools (interview schedule and observation checklist) were used for data collection, which was designed after being reviewed by and discussed with expert's committees comprising veterinarians, public health experts medical officers, experts in health education and statistician as a Delphi technique [14] which made research tools more realistic and then tools were pre-tested. The questionnaire is divided into four sections: (i) respondents' socio-demographic status; (ii) knowledge about zoonoses; (iii) zoonoses preventive practice; and (iv) perception measuring questionnaire (however, based on objectives we only picked knowledge and practice related data and excluded the perception related part in this article). According to the nature of the data, collected data were entered in to the Microsoft excel sheet. After completing the data entry works 10 percent of the total sample (n = 380) was re-checked and then data were imported to SPSS software. The data were calculated according to the nature of the variables by using descriptive statistics. Univariate and bivariate analysis were performed. Univariate analysis such as frequencies and percentage were used to describe the characteristics of respondents. Bivariant analyses such as the chi-square test were used to examine the association between independent and dependent variables.

### Results

This study was conducted in Manang, Tanahun, and Nawalpur districts which are located in Gandaki Province and represent all three ecological regions (Mountain, Hill, and Terai) of Nepal. The altitude of the terai region is from 60 to 610 meters, whereas the altitude of the hilly region is from 610 to 4877 and the altitude of the mountain region is from 4877 meters to 8848 meters from the sea level [15]. Among the respondents, 32 (8.4%) were from Manang, 176

**Table 1. Socio-demographic characteristics of the respondents.**

| Variables | Attributes | Frequency | Percent |
|---|---|---|---|
| District | Manang | 32 | 8.4 |
| | Tanahun | 176 | 46.3 |
| | Nawalpur | 172 | 45.2 |
| Gender | Female | 174 | 45.8 |
| | Male | 206 | 54.2 |
| Age group | 20–39 | 227 | 59.7 |
| | 40–59 | 128 | 33.7 |
| | 60 and above | 25 | 6.6 |
| Main occupation of household | Livestock/ Poultry | 30 | 7.9 |
| | Agro farming | 257 | 67.6 |
| | Government service | 33 | 8.7 |
| | Foreign employee | 29 | 7.6 |
| | Trade | 27 | 7.1 |
| | Other | 4 | 1.1 |
| Educational status of respondents | Illiterate | 16 | 4.2 |
| | Up to basic level | 199 | 52.4 |
| | Secondary and above | 165 | 43.4 |
| Aim of livestock farming | Household consuming | 328 | 86.3 |
| | Commercial farming | 52 | 13.7 |
| Training related to farming | Yes (short course) | 11 | 2.9 |
| | No | 369 | 97.1 |
| Type of livestock farming | Single | 31 | 8.2 |
| | Mixed farming | 349 | 91.8 |
| Keeping livestock in the household + | Cow | 107 | 28.2 |
| | Buffalo | 200 | 52.6 |
| | Goat/sheep | 260 | 68.4 |
| | Pig | 26 | 7.1 |
| | Poultry | 340 | 89.5 |
| | Yak/Chauri | 7 | 1.8 |
| | Other | 8 | 2.1 |

Note: + Percentage exceeds 100 due to multiple responses

(46.3%) were from Tanahun, and 172 (45.2%) were covered by Nawalpur district as probability proportional to size (PPS) sampling technique.

Table 1 shows that majority (54.2%) of the respondents in the study were male, nearly 3 in 5 (59.7%) were between 20 to 39 years and the median age of the respondents was 35 years. More than two-thirds (68%) of respondents adopted agro-farming and 7.9 percent were dependent on livestock farming as the main occupation in their households. More than half (52.0%) had basic level education, and very nominal (3.0%) of livestock farmers got training related to livestock farming. Likewise, most of the respondents (91.8%) were keeping mixed types of livestock in their household, where poultry, goat/ sheep, and buffalo are highly keeping livestock (90%, 68%, and 53% respectively) in their household and the main purpose of that farming was household consumption (86%) in the study.

### Knowledge of common zoonoses based on ecological regions

**Knowledge of zoonotic rabies in the study.**    Rabies is a viral zoonotic disease. Most often it is transmitted through the bite of a rabid animal. Most of the respondents (91%, n = 380) had heard about zoonotic rabies. However, higher respondents in Nawalpur (95.3%, n = 172) had known about zoonotic rabies than in the other two districts which seem significant association ($\rho$ = 0.007) between zoonotic rabies-related knowledge based on the different ecological regions in the study.

Table 2 also shows that, among the total respondents, one-third (33.3%) were known about the causative agent of rabies with higher respondents in Nawalpur (53.7%, n = 172) than in Manang (10.0%, n = 32) and Tanahun (16.0%, n = 176). However, respondents had good knowledge of the symptoms of rabies (hydrophobia) by 98.0 and 87.2 percent respondents in Tanahun and Nawalpur than the symptoms of salivation (11.3% and 66.5%) and irritation (8.6% and 23.2%) respectively. But, in Manang greater number of the respondents knew about symptoms of salivation (63.3%) than hydrophobia and irritation (36.7% and 26.7%).

Bitten by rabid animals or contact saliva in an open wound is the mode of transmission of the rabies virus [16]. Knowledge of these variables was found satisfactory results in all study districts. Where, 90.0, 91.1, and 95.1 percent of respondents were known in Manang, Tanahun, and Nawalpur districts respectively and similar knowledge was also found in the preventive measures of rabies in all study districts. Where higher respondents in Nawalpur had a piece of knowledge of preventive methods (99.4%) to compare the Tanahun (98%) and Manang (83%). However, knowledge of the free availability of anti-rabies vaccine (ARV) up to primary health

**Table 2. Rabies; knowledge-based on ecological regions of the respondents.**

| Variables | Total | | Manang | | Tanahun | | Nawalpur | | P-value |
|---|---|---|---|---|---|---|---|---|---|
| | N = 380 | % | N = 32 | % | N = 176 | % | N = 172 | % | |
| Heard | 345 | 90.7 | 30 | 93.8 | 151 | 85.8 | 164 | 95.3 | .007* |
| No heard | 35 | 9.2 | 2 | 6.2 | 25 | 14.2 | 8 | 4.7 | |
| Causative agent of rabies | | | | | | | | | |
| Virus# | 115 | 33.3 | 3 | 10.0 | 24 | 15.9 | 88 | 53.7 | .000* |
| Others | 91 | 26.3 | 2 | 6.7 | 76 | 50.3 | 13 | 7.9 | |
| I don't know | 139 | 40.2 | 25 | 83.3 | 51 | 33.8 | 63 | 38.4 | |
| Symptoms of rabies + | | | | | | | | | |
| Hydrophobia | 302 | 87.5 | 11 | 36.7 | 148 | 98.0 | 143 | 87.2 | .000* |
| Salivation | 145 | 40.0 | 19 | 63.3 | 17 | 11.3 | 109 | 66.5 | .000* |
| Irritation | 59 | 17.1 | 8 | 26.7 | 13 | 8.6 | 38 | 23.2 | .000* |
| Rabies mode of transmission | | | | | | | | | |
| Bitten by rabid animals. . . # | 319 | 92.4 | 27 | 90.0 | 136 | 90.1 | 156 | 95.1 | .000* |
| Other | 15 | 4.3 | 0 | 0 | 15 | 9.9 | 0 | 0 | |
| I don't know | 11 | 12.1 | 3 | 10.0 | 0 | 0.0 | 8 | 4.9 | |
| Preventive measure of rabies | | | | | | | | | |
| Wash the wound and contact Hospital # | 336 | 97.4 | 25 | 83.3 | 148 | 98.0 | 163 | 99.4 | .000* |
| Others | 4 | 1.1 | 1 | 3.3 | 3 | 2.0 | 0 | 0 | |
| I don't know | 5 | 1.4 | 4 | 13.3 | 0 | 0.0 | 1 | 0.6 | |
| Knowledge of anti-rabies vaccine (ARV) freely availability in Government Hospital | | | | | | | | | |
| I know | 153 | 44.3 | 0 | 0 | 40 | 26.5 | 113 | 68.9 | .000* |
| I don't know | 192 | 55.7 | 30 | 100. | 111 | 73.5 | 51 | 31.1 | |

Note: + Percentage exceeds 100 due to multiple responses # correct answer * chi-square value

**Table 3. Bird flu; knowledge based on ecological regions of the respondents.**

| Variables | Total | | Manang | | Tanahun | | Nawalpur | | p-value |
|---|---|---|---|---|---|---|---|---|---|
| | N = 380 | % | N = 32 | % | N = 176 | % | N = 172 | % | |
| Heard about Bird flu | 364 | 95.8 | 28 | 87.5 | 173 | 98.3 | 163 | 94.8 | .013* |
| No Heard | 16 | 4.2 | 4 | 12.5 | 3 | 1.7 | 9 | 5.2 | |
| Causative Agent of Bird flu | | | | | | | | | |
| Virus# | 226 | 62.1 | 9 | 32.1 | 123 | 71.1 | 94 | 57.7 | .000* |
| Others | 11 | 3.0 | 0 | 0 | 5 | 2.9 | 6 | 3.7 | |
| I don't know | 127 | 34.9 | 19 | 67.9 | 45 | 26.0 | 63 | 38.7 | |
| Bird flu Mode of Transmission + | | | | | | | | | |
| Contact without safety. . .. | 283 | 77.7 | 12 | 42.9 | 124 | 71.7 | 147 | 90.2 | .000* |
| Consumed raw poultry product | 349 | 95.9 | 25 | 89.3 | 171 | 98.8 | 153 | 93.9 | .000* |
| Contact infected poultry | 81 | 22.2 | 12 | 42.9 | 21 | 12.1 | 48 | 29.4 | .000* |
| Symptoms of Bird flu in Poultry + | | | | | | | | | |
| Diarrhea | 124 | 34.1 | 3 | 10.7 | 105 | 60.7 | 16 | 9.8 | .000* |
| Nasal discharge | 51 | 14.0 | 3 | 10.7 | 34 | 19.7 | 14 | 8.6 | .027* |
| Cough sneezing | 21 | 5.8 | 2 | 7.1 | 13 | 7.5 | 6 | 3.7 | .303* |
| Swelling eyelid and joint. . . . . . | 224 | 61.5 | 3 | 10.7 | 159 | 91.9 | 62 | 38.0 | .000* |
| Preventive Measures of Bird flu + | | | | | | | | | |
| Consumed well-cooked poultry. . . | 345 | 94.8 | 22 | 78.6 | 172 | 99.4 | 151 | 92.6 | .000* |
| Handwashing with soap | 286 | 78.6 | 22 | 78.6 | 143 | 82.7 | 122 | 74.8 | .141* |
| Use PPE | 153 | 42.1 | 7 | 25.0 | 40 | 23.1 | 106 | 65.0 | .000* |

Note: + Percentage exceeds 100 due to multiple response # correct response * chi-square value

center (PHC) found poor knowledge (0 and 26.5%) in Manang, and Tanahun respectively than in Nawalpur (68.9%), and all variables related to knowledge found significant association ($p<0.001$) with the ecological regions in the study.

**Knowledge of bird flu in the study.** Bird flu is caused by the influenza type A (H5N1) virus. H5N1 virus naturally occurs among wild aquatic birds and can infect domestic poultry and other birds and animal species. Normally, it does not infect humans, however infection may occur during inhalation if the virus is present in ambient air [3]. It is a well-known zoonotic disease among the respondents and is also highly infectious in avian. However, due to close exposure without any safety, people may get bird flu from their poultry which may have a fatal outcome.

Table 3 shows that, most of the respondents in the study (96%) were heard about zoonotic bird flu. However, to compare the studies districts; higher respondents in Tanahun had a higher knowledge of bird flu (98.3%) than in Nawalpur (94.8%), and Manang (87.5%). But, knowledge of causative agents was found lower to compare the previous variable (hearing about bird flu) in the study. Ecologically, only 71, 58, and 32 percent of respondents knew about the causative agents of bird flu with higher variation in Tanahun, Nawalpur and Manang respectively with statistically significant ($p<0.001$).

If people have information about the way of disease transmission, they become capable to break this chain which in turn helps to protect from infection. In the case of bird flu, results were found satisfactory in the study. Respondents in all study districts had a higher knowledge of bird flu; it is transmitted due to consuming raw poultry products (meat or egg) with regional variation ($p<0.001$). However, knowledge of symptoms of bird flu in poultry was found poor among the respondents in Manang (7.1–10.7%) and Nawalpur (3.7–38.0%) compare to the Tanahun (7.5–91.9%) with statistically significant ($p <0.001$ to .027) in the study. Similarly,

respondents had a good knowledge of preventive measures for bird flu in Tanahun in comparison to the other two districts.

**Knowledge of swine flu in the study.** It is a highly contagious zoonotic disease transmitted by an infected pig (swine). People who are close to infected pigs without safety will be vulnerable to swine flu. Thus, farmers, who have a swine farm most have a basic knowledge of the pig-related zoonoses.

Table 4 also shows that, more than half (54%, n = 380) of respondents have heard about swine flu. However, a significant number of respondents had unknown about it. Ecologically, higher respondents in Nawalpur (61.6%) knew about swine flu than in Tanahun (52.8%) and Manang (21.9%) with significant regional variation ($\rho < 0.001$). Similarly, more than half (58.7%, n = 206) of respondents knew about the causative agent of zoonotic swine flu. Ecologically, more respondents (72%) in Tanahun than in Nawalpur (48%) and in Manang (43%) had knowledge of the causative agent ($\rho < 0.001$).

Variables on symptoms and mode of transmission of swine flu were found having similar knowledge of the farmers in the study. However, the highest number of respondents (95.7%) in Tanahun knew about symptoms of swine flu, to compare the respondents of Nawalpur (63.2%) and Manang (57.1%) with significant regional variation ($\rho < 0.001$). Mask wearing, one of the easy and cost-effective preventive techniques for swine flu, also has quite similar practices in all study districts. Whereas don't touch facial part unnecessarily is responded by higher respondents in Tanahun (95%) than in Manang (71%) and Nawalpur districts (63%, $\rho < 0.001$). However, avoiding unnecessary travel was responded by the least respondents in the study. Out of the studied six common zoonotic diseases, very few of the respondents knew about zoonotic brucellosis (1.6%), neurocysticercosis (2.7%), and bovine tuberculosis (3.1%), so we present data very briefly in the study.

**Table 4. Swine flu; knowledge based on ecological regions of the respondents.**

| Variables | Total | | Manang | | Tanahun | | Nawalpur | | p-value |
|---|---|---|---|---|---|---|---|---|---|
| | N = 380 | % | N = 32 | % | N = 176 | % | N = 172 | % | |
| Heard | 206 | 54.2 | 7 | 21.9 | 93 | 52.8 | 106 | 61.6 | .000* |
| No heard | 174 | 45.8 | 25 | 78.1 | 83 | 47.2 | 66 | 38.4 | |
| Causative agent of swine flu | | | | | | | | | |
| Virus# | 121 | 58.7 | 3 | 42.9 | 67 | 72.0 | 51 | 48.1 | .000* |
| Others | 19 | 9.2 | 0 | 0 | 16 | 17.2 | 3 | 2.8 | |
| I don't know | 66 | 32.1 | 4 | 57.1 | 10 | 10.8 | 52 | 49.1 | |
| Symptoms of swine flu | | | | | | | | | |
| Like a normal flu | 160 | 77.7 | 4 | 57.1 | 89 | 95.7 | 67 | 63.2 | .000* |
| Others | 3 | 1.4 | 0 | 0 | 1 | 1.1 | 2 | 2.1 | |
| I don't know | 43 | 20.9 | 3 | 42.9 | 3 | 3.2 | 37 | 34.9 | |
| Swine flu mode of transmission | | | | | | | | | |
| Droplet infection | 160 | 77.7 | 4 | 57.1 | 89 | 95.7 | 67 | 63.2 | .000* |
| Other | 2 | 0.9 | 0 | 0.0 | 1 | 1.1 | 1 | 0.9 | |
| I don't know | 44 | 21.3 | 3 | 42.9 | 3 | 3.2 | 38 | 35.8 | |
| Preventive measures of swine flu + | | | | | | | | | |
| Don't touch the facial part | 160 | 77.7 | 5 | 71.4 | 88 | 94.6 | 67 | 63.2 | .001* |
| Avoid travel unnecessarily | 36 | 17.5 | 3 | 42.9 | 10 | 10.8 | 23 | 21.7 | .050* |
| wear mask | 123 | 59.7 | 5 | 71.4 | 55 | 59.1 | 63 | 59.4 | .060* |

Note: + Percentage exceeds 100 due to multiple response # correct response* chi-square value

**Knowledge of brucellosis in the study.**    Brucellosis is a common zoonotic disease in Nepal [17]. It is transmitted commonly from aborted livestock and contaminated milk and milk production. However, very least (1.6%, n = 380) respondents knew about zoonotic brucellosis in the study. Ecologically, no one respondents in Manang and equally (1.8%, n = 176) in Tanahun and Nawalur (1.8%, n = 172) heard about brucellosis. Therefore, the respondents in all study districts had very low knowledge of brucellosis.

**Knowledge of neurocysticercosis (NCC) in the study.**    It is a zoonotic disease or condition which occurs when a cyst of tapeworm travels into the human brain. It is a common zoonosis in Nepal. However, very few respondents (2.7%, n = 380) in the study knew about NCC. Ecologically, higher respondents in Manang (13%, n = 32) than in Tananhu (1.7%, n = 176) and Nawalpur (1.8%, n = 172) heard about NCC with significant regional variation ($\rho < 0.001$).

**Knowledge of bovine tuberculosis in the study.**    Bovine tuberculosis is also a common zoonotic disease in Nepal [13]. It is a disease caused by the bacterium Mycobacterium bovis and can infect many species of animals particularly cattle and buffalo and can spread to human [18]. People living in close relation with cattle without any protective measures are considered to be at risk of bovine tuberculosis, however, very few respondents (3.1%, n = 380) had knowledge about it. To compare the study districts, a higher number of respondents in Nawalpur (5.2%, n = 172) heard about bovine tuberculosis than those in Manang (0.0) and Tanahun (1.7%, n = 176).

## Zoonoses preventive practices based on ecological regions

Because of the host and reservoir characteristics of animals, livestock farmers who have close exposure to their livestock are facing threats to zoonoses. Therefore, to prevent from zoonoses, they need to follow safety practices during close exposure to their livestock. However, respondents have poor preventive practices in the study. Regarding hand washing practices, only a little more than half (60.8%, n = 380) of the respondents washed their hands with soap water regularly after close exposure to their livestock. Ecologically, higher respondents (91.9%, n = 172) in Nawalpur regularly followed this practice than in Manang (71.9%, n = 32) and Tanahun (28.4%. n = 176). Mask-wearing practices during close exposure were also found to be poor practices in the study, where only 25 (6.6%) livestock farmers wore masks regularly. To compare the results in study districts, higher number of respondents in Nawalpur followed the regular mask using practices (12%, $\rho < 0.001$), with significant regional variation (Table 5).

Table 5 shows that, only 7 (1.8%) respondents used gloves regularly. Ecologically, respondents in Manang were found to be higher regular gloves users (3.1%) than those in the other two districts during close exposure to livestock. Likewise, boot-wearing practices resembled the glove wearing practices (1.3%, n = 380) among the respondents with the higher number of regular users in Manang (6.3%, n = 32), with significant regional variation ($\rho < 0.001$). Those practices indicate that livestock farmers are facing zoonoses-related vulnerability due to a lack of properly used PPE. However, the proportion between using practices and availability of PPE in the household (like; soap, mask, gloves, and boots) was found to be compatible during the interview period.

Most of the respondents were found to be keeping mixed types of livestock in their households with traditional practices. Economic status, education, or several socio-cultural values might determine farm and household sanitation in farming communities. Only a few of the respondents (12.1%, n = 380) maintain a standard distance between the home and the shed in the study (Table 6). From the perspective of public health, the normal standard distance between home and shed is more than 15 meters or 50 fits [19]. Based on ecological region, most respondents in Manang (91%, n = 32) maintain this standard in comparison to those in

**Table 5. Zoonoses preventive practices based on the ecological region of the respondents.**

| Variables | Frequencies | | Manang | | Tanahun | | Nawalpur | | P-value |
|---|---|---|---|---|---|---|---|---|---|
| Handwashing with soap water after close contact with livestock | | | | | | | | | |
| | N = 380 | % | N = 32 | % | N = 176 | % | N = 172 | % | |
| Regular | 231 | 60.8 | 23 | 71.9 | 50 | 28.4 | 158 | 91.9 | .564* |
| Occasionally | 149 | 39.2 | 9 | 28.1 | 126 | 71.6 | 14 | 8.1 | |
| Mask wearing during exposure to livestock/poultry | | | | | | | | | |
| Regular | 25 | 6.6 | 2 | 6.3 | 2 | 1.1 | 21 | 12.2 | .000* |
| Occasionally | 161 | 42.4 | 4 | 12.5 | 107 | 60.8 | 50 | 29.1 | |
| Never | 194 | 51.1 | 26 | 81.3 | 67 | 38.1 | 101 | 58.7 | |
| Gloves used during caring the livestock/poultry | | | | | | | | | |
| Regular | 7 | 1.8 | 1 | 3.1 | 1 | 0.6 | 5 | 2.9 | .000* |
| Occasionally | 106 | 27.9 | 15 | 46.9 | 71 | 40.3 | 20 | 11.6 | |
| Never | 267 | 70.3 | 16 | 50.0 | 104 | 59.1 | 147 | 85.5 | |
| Boot wearing during exposure to livestock/poultry | | | | | | | | | |
| Regular | 5 | 1.3 | 2 | 6.3 | 0 | 0 | 3 | 1.7 | .000* |
| Occasionally | 77 | 20.3 | 19 | 59.4 | 46 | 26.1 | 12 | 7.0 | |
| Never | 298 | 78.4 | 11 | 34.4 | 130 | 73.9 | 157 | 91.3 | |
| Availability of the above PPE in households during interview time | | | | | | | | | |
| Soap | 347 | 91.3 | 32 | 100 | 167 | 94.9 | 148 | 86.0 | |
| Mask | 150 | 39.4 | 5 | 15.6 | 64 | 36.4 | 81 | 47.1 | |
| Gloves | 56 | 14.8 | 10 | 31.3 | 19 | 10.8 | 27 | 15.7 | |
| Boots | 31 | 8.1 | 20 | 62.5 | 5 | 2.8 | 6 | 3 | |

Note: * = Chi-square test

Nawalpur (9%, n = 172) and Tanahun (1%, n = 176), with significant regional variation ($\rho<0.001$).

Table 6 also shows that nearly 9 out of 10 respondents (88%) dispose dead animal on bore hole, however, reversed is true in case of Manang where 81% respondents through dead animals in the nearest river. Similarly, more than two-thirds (67%) of respondents said that their children had close exposure to livestock. Those exposures were found higher in Manang and Tanahun than in Nawalpur (88, 85, and 45%) respectively. Likewise, nearly two-thirds (65.3%) of households' women had close exposure for care taker to their livestock during their pregnancy period. This practice was found higher in Manang (97%) and Tanahun (87%) than in Nawalpur (37%). Likewise, 64 (17%) households still consumed the meat of sick animals. This practice was found higher in Manang (72%) than in Nawalpur (14%) and Tanahun (10%). Vaccination practices for their livestock were also found to have poor coverage in the study. Out of the total respondents, only two-thirds (36%) vaccinated their livestock. To compare this practice, Nawalpur had the highest coverage (72%) in comparison to Manang (9%) and Tanahun (5%). Based on the data, the majority of the farmers were found to have ignored vaccination to livestock as pre-exposure prophylaxis in all ecological zones, which may induce zoonoses vulnerabilities to the farming communities and all preventative practices were found to have significant regional variation ($\rho<0.001$) in the study.

## Discussion

In Nepal as an agrarian country, a large number of people are involved in the field of agro or livestock farming [2], and farmers are exposed to zoonotic agents in every aspect of their work.

**Table 6. Livestock related common practices based on ecological regions of the respondents'.**

| Variables | Frequencies | | Manang | | Tanahun | | Nawalpur | | P-value |
|---|---|---|---|---|---|---|---|---|---|
| | N = 380 | % | N = 32 | % | N = 176 | % | N = 172 | % | |
| Distance between Home and Shed | | | | | | | | | |
| Less than 15 M. | 334 | 87.9 | 3 | 9.4 | 174 | 98.9 | 157 | 91.3 | .000* |
| More than 15 M. | 46 | 12.1 | 29 | 90.6 | 2 | 1.1 | 15 | 8.7 | |
| Disposal practices of dead animals | | | | | | | | | |
| Borehole | 336 | 88.4 | 6 | 18.8 | 176 | 100. | 154 | 89.5 | .000* |
| Throw the river | 44 | 11.6 | 26 | 81.2 | 0 | 0 | 18 | 10.5 | |
| Children exposure to livestock | | | | | | | | | |
| Yes | 254 | 66.8 | 28 | 87.5 | 149 | 84.7 | 77 | 44.8 | .000* |
| No | 126 | 33.2 | 4 | 12.5 | 27 | 15.3 | 95 | 55.2 | |
| Pregnant women exposure to livestock | | | | | | | | | |
| Yes | 248 | 65.3 | 31 | 96.9 | 153 | 86.9 | 64 | 37.2 | .000* |
| No | 132 | 34.7 | 1 | 3.1 | 23 | 13.1 | 108 | 62.8 | |
| Sick animal consuming practices | | | | | | | | | |
| Yes | 64 | 16.8 | 23 | 71.9 | 17 | 9.7 | 24 | 14.0 | .000* |
| No | 316 | 83.2 | 9 | 28.1 | 159 | 90.3 | 148 | 86.0 | |
| Vaccination practice for livestock | | | | | | | | | |
| Yes | 135 | 35.5 | 3 | 9.4 | 8 | 4.6 | 124 | 72.1 | .000* |
| No | 245 | 64.5 | 29 | 90.6 | 168 | 95.4 | 48 | 27.9 | |

Note: M = meter // * = chi-square test

It was found that many zoonotic diseases are prevalent in Nepal in various forms (sporadic, endemic, epidemic, and so on) [7]. Thus, people who have close exposure to livestock or related professions need to be knowledgeable about livestock-related common zoonoses which are prevalent in their territory.

In this study, we found that avian influenza was the most commonly known zoonotic disease among the respondents, followed by rabies and swine flu. However, the respondents had poor knowledge of bovine tuberculosis, neurocysticercosis, and brucellosis, although all studied zoonotic diseases have endemic potential in Nepal [13] and have a direct impact on the health of livestock and caretaker farmers. Because of identical clinical manifestations with common mild infections, people are more likely to ignore zoonotic diseases including brucellosis, bovine tuberculosis, and neurocysticercosis.

With knowledge compared to studied zoonoses among the livestock farmers in all three ecological regions, it was found that higher respondents in Nawalpur knew about zoonotic rabies, swine flu, bovine tuberculosis, and brucellosis. Likewise, respondents in Tanahun were found to have higher knowledge of avian influenza and so was with neurocysticercosis in Manang.

Rabies is a life-threatening zoonotic disease that has endemic potential in many parts of developing countries like Nepal. Due to close contact with livestock without any safety and the burden of stray dogs, almost half of the Nepalese population is at risk, and a quarter are at moderate risk of rabies [20]. Most respondents in the study heard about zoonotic rabies with regional variation, which was higher than among livestock farmers in Punjab, India [21].

With results compared among the study districts on zoonotic rabies, more respondents in Nawalpur responded with the correct answer on causative agents, mode of transmission, preventive measures, and free ARV services (it is supply based on case load and institutional demand system) up to the PHC and district level government hospital in Nepal [22]. However,

more respondents in Tanahun responded with the correct answers about the symptoms of rabies. Those results were found to be similar to livestock farmers in Punjab, India [21] and traditional farmers in Jimma, Southwestern Ethiopia, where (82.8%; n = 75) respondents perceived the prognosis of rabies as that of a person who was bitten by a rabid dog and had not gotten ARV would die [23] which was higher knowledge than in this study.

As a health educator, when we analyzed the data with a critical perspective, rabies is a fatal disease if the victim does not take an ARV as post-exposure prophylaxis (PEP). However, knowledge of ARV was not found satisfactory in this study. Due to poor knowledge of disease fatalities and free ARV services, people may not perceive disease threats and refuse ARV, which contributes to the rabies burden in the Nepalese community and also creates a challenge to achieving the ending dog mediated human rabies deaths by 2030 in Nepal [22].

Similarly, due to the frequent outbreaks avian influenza was mostly known zoonotic disease among the respondents. In Nepal, avian influenza was first time seen in 2009, after that it was seen 237 outbreaks were reported in different parts of the country till fiscal years 2016/17 and 1,966,745 poultry were slaughtered [24]. However, more respondents in Tanahun responded with the correct answer about the diseases in comparison to the other two districts. This result was similar to livestock farmers in Punjab, India [21] where 92.4 percent of farmers knew about avian influenza.

Swine flu is a highly contagious disease with a pandemic nature. With the first outbreak in Mexico in 2009, 213 countries globally reported laboratory confirmed cases till 2010 resulting in 16455 deaths. In Nepal, 172 case were reported with 2 deaths in 2010 [25] and it is listed among common zoonotic disease [13] and shows the endemic distribution. In this study, only more than half of the respondents had heard about swine flu, which indicates vulnerability to zoonotic swine flu in farming communities. Data comparing the study's three districts, more respondents in Nawalpur had knowledge of swine flu than in Tanahun and Manang. This result was very low in comparison to the study in Punjab, India, where almost all respondents heard about swine flu [21].

Similarly, very few respondents (3.1%) heard about bovine tuberculosis, with higher respondents in Nawalpur responding with the correct answer than in the other two districts. However, the percentage of the correct responses in all variables was found very low in all study districts. It was poor knowledge of the livestock farmers in comparison to the farmers in Punjab, India [21], smallholder dairy farmers in North Shewa, Ethiopia [26], and community people in Ghana [27].

Bovine tuberculosis is a cattle-related common zoonosis, but most livestock farmers in Nepal have little or no knowledge of bovine tuberculosis, which poses a challenge not only for veterinarians but also for public health and community members. A study conducted in Chitwan, Nepal, it was found that in 60 tuberculosis patients' households, 15 percent of livestock were found to have bovine tuberculosis, and all tuberculosis patients were involved in feeding, milking, and other roles of taking care of livestock, and one-fourth of patients' habits of raw milk consumption [28]. Those types of practices are vulnerable to further disease transmission in the communities. Likewise, in households with 50 cattle, after the intradermal cervical tuberculin test, 10 percent of animals were found to have Mycobacterium bovis [29]. Due to a lack of proper study, we had no data on the proportion of human tuberculosis cases from bovine sources. However, it is believed that most of extra pulmonary (EP) tuberculosis cases in humans are from bovine sources [30].

Brucellosis is the second most important disease in the world after rabies, and it has been reported as endemic in Nepal [17]. Respondents had the least knowledge about brucellosis, with only 1.6 percent having heard about it, which was very low in comparison to the finding [31] in Tajikistan, where 15.0 percent of respondents had heard about brucellosis, and a similar

finding was there in Shewa, Ethiopia [26], but none of the respondents had heard of such a disease as brucellosis in South-western Ethiopia [23]. In comparison, 88 percent of farmers in Kars, Turkey, were aware of brucellosis [32] and livestock farmers in Punjab were also aware of the disease [21]. This study also proved that brucellosis-related knowledge in respondents seems associated with the economic and educational status of the countries, which was proved when comparing the studies on Punjab, India, Kars, Turkey, Ethiopia, and Nepal.

In the same way, the knowledge of neurocysticercosis was low (2.7%) among the respondents when compared to the study in Southwestern Ethiopia, where two-thirds of the traditional farmers (n = 48) knew taeniasis. They also shared that taeniasis is transmitted to humans when raw meat [Taenia saginata (beef tapeworm), Taenia solium (pork tapeworm)] is consumed [23], but more than two-thirds of traditional farmers had a habit of raw meat consumption, and among the raw meat consumers, almost all have been infected by taenia saginata at least once previously. Several patient reports from Asian countries indicate the wide prevalence of taenia solium cysticercosis, and it is the major cause of epilepsy in Indonesia, Vietnam, and Nepal [33]. When compared with the districts, respondents in Manang were found to furnish higher correct answers in all knowledge measuring questionnaires (3.1–12.5%) than in other districts.

Human behaviors and practices are influenced by their existing knowledge, socio-cultural values, or perceptions in a particular circumstance. In the context of Nepal, most livestock farmers follow traditional practices, adopting mixed types of livestock farming and exposing themselves to threats. However, due to the host and reservoir characteristics of several lethal pathogens, caretaker farmers are facing susceptibilities to zoonotic diseases by their keeping livestock.

Regarding hand washing practice, more than sixty percent of respondents in this study wash their hands with soap and water on a regular basis after close exposure to livestock, and similar practices were found in all ecological regions in the study. In contrast, in a study in Chitwan, Gorkha, and Tanahun districts, a higher number (94.0%) of smallholder farmers wash their hands with soap water after handling livestock [34], and in a study in the suburban areas of Bangladesh, where (100.0%) of smallholder livestock farmers wash their hands with soap water after interaction with animals [35].

Mask wearing during close exposure to livestock was also found to be poor practice in the study. Ecologically, the very worst scenario was found in Tanahun (1%) in comparison to Manang (6%) and Nawalpur (12%), and this finding was similar to farmers in Kars, Turkey [32], where equal respondents as in this study (6.6%) used masks regularly, but 84% of farmers considered it necessary. Not only that, but also gloves wearing during close exposure to their livestock was also found to be poor practices (1.8%) in the study. This was a poor practice in comparison to the study in Kars, Turkey [32], where 35.8% of farmers used gloves regularly while 92.1% had a positive attitude. But, only a few (0.8%) cattle farmers in the Tamale, northern region of Ghana, used gloves while handling sick cattle [36]. Likewise, boot-wearing practices in regular basis were also found poor practices (1.3%), than in farmers in Kars, Turkey [32], where 42.4 percent of respondents wore boots on a regular basis during the exposure period, with 89.4 percent positive perception. Therefore, the poor practice of most cost effective personal protective measures, i.e., hand washing, mask, gloves and boot wearing poses a remarkable threat to zoonoses with significant ecological variation ($\rho < 0.001$) in the study.

This study revealed many risk factors related to conventional livestock farming practices in Nepal. Economic status, education, or several socio-cultural values determine farm and household sanitation in farming communities. However, only 12.3 percent of respondents were found to have maintained a standard distance between home and shed in the study, and similar practice was observed in Bangladesh [37]. But in another study in Bangladesh, nearly half

(43.0%) of the farms were attached to their houses [35]. From the public health point of view, the standard distance between a house and a shed is more than 15 meters or 50 feet [19].

Several zoonotic diseases like rabies, avian influenza, swine flu, neurocysticercosis, brucellosis, bovine tuberculosis, etc. are considered to be potential zoonotic diseases in Nepal [13]. Along with these diseases, several other zoonotic diseases exist that will be transmitted to humans mainly through direct contact during care-taking or other close exposure without PPE to the infected animal. Twelve percent of respondents in the study still throw dead livestock into the nearest river, which was found to be similar practices on smallholder farmers in other parts of Nepal [34] where 18 percent of farmers disposed of animal waste (like; placenta) on the ground, in water, or on a tree as a customary practice, which was vulnerable to zoonotic transmission. Ecologically, higher respondents in Manang adopted these practices which found a significant association with the ecological region and dead animal disposal practices in the study (p = 0.001).

There were several influencing factors for close exposure to children and livestock. Either to minimize the parent's workload or as a normal refreshment activity, the children of the respondents were close to livestock in this study. More than two-thirds of respondents in the study disclosed that their children had close exposure to livestock. Ecologically, in Nawalpur more respondents avoided those practices (55.2%) than in Tanahun (15.3%) and Manang (12.5%), with significant regional variation ($\rho < 0.001$). Not only in Nepal, but similar results (70.0%) were revealed in suburban areas of Bangladesh, where children of respondent farmers had close exposure to their animals [35]. This may impact the education and economic status of the livestock farmers, who are found similarly in Bangladesh and Nepal.

Not only the children, but nearly two-thirds (65%) of the respondents stated that pregnant women were also close to their livestock as caregivers during their pregnancy. These practices were found higher in Manang than in other districts with significant regional variation in the study ($\rho < 0.001$). Due to unstable immunity, those types of exposure may create a higher vulnerability to the health of the mother, fetus, or younger children from zoonotic diseases [38].

The consumption practices of sick or recently dead animals are influenced by several socioeconomic statuses. However, some of the respondents in this study (17%) still follows those practicing in their communities. These practices were found to be more prevalent in Manang (72%), than in Tanahun (14%), and Nawalpur (9.7%), with a significant ecological association ($\rho < 0.001$). In contrast, good knowledge was related to sick and dead animals, where four out of five respondents knew that they should be buried deep and practiced by one-fifth in Turkey [32].

Most communicable diseases (including zoonoses from livestock) will be controlled by a cost-effective single-dose vaccination. However, due to several obstacles, farmers are facing many losses in the field of livestock farming. In this study, vaccination practices were also found to have poor coverage among the respondents. Only two-thirds of respondents vaccinated their livestock with pre-exposure prophylaxis, which was poorer practice than in suburban areas of Bangladesh (78.26%) [35], which might be responsible for zoonotic outbreaks anytime and anywhere.

## Limitations and recommendations for future research

Using the cross-sectional survey research design in this study does not capture changes over time, only representing one time-slice of data. Due to several influencing factors such as time and budget, out of the listed eleven common zoonoses in Nepal [13], researchers covered only six which are near to livestock. Infectious diseases do cross geographic boundaries, as we have learned from recent pandemics (such as COVID-19) where serious diseases spread across countries. Future research should be designed to both extend and replicate data collected in

this study concerning zoonotic health beliefs and behaviors by collecting data across countries to provide a broader understanding of international zoonoses prevention patterns.

While the use of the quantitative survey data collection method in this study provided important insights into the knowledge and preventive practices related to zoonoses of Nepalese livestock farmers. To capture the changing practices over time, a longitudinal study will be helpful. Future studies could build upon the findings from probing qualitative interviews, in order to more deeply examine the variety of social, economic, cultural, and environmental factors that are most influential in the adoption of relevant zoonotic disease prevention guidelines.

## Conclusion

Majority of the people in Nepal are involved in the farming profession, and they are exposed to zoonotic agents in every aspect of their work. So, to prevent zoonoses, they must have the knowledge and skills to practice preventive measures. However, the majority of respondents have an insufficient idea about many of the studied zoonotic diseases that are endemic in their territory, and we found a significant regional variation in knowledge and zoonotic preventive practices in the study. Therefore, if we want to make the livestock farming profession safer from the perspective of zoonoses, we should plan the interventional programs for livestock farmers based on the One Health approach, which might not only contribute to healthy livestock and farming communities but also equally support to eradicate poverty and increase the national GDP.

## Supporting information

**S1 Data. Supplemental data.**
(SAV)

## Acknowledgments

This study was a part of the PhD of the first author. We would like to thank the participants for their time and responses. We would also like to thank Mr. Suresh Shrestha and Mr. Dinesh Panthee for their valuable contribution for editing the manuscript.

## Author Contributions

**Conceptualization:** Kosh Bilash Bagale, Ramesh Adhikari, Devaraj Acharya.

**Data curation:** Kosh Bilash Bagale.

**Formal analysis:** Kosh Bilash Bagale, Ramesh Adhikari, Devaraj Acharya.

**Methodology:** Kosh Bilash Bagale, Ramesh Adhikari, Devaraj Acharya.

**Software:** Devaraj Acharya.

**Supervision:** Ramesh Adhikari, Devaraj Acharya.

**Validation:** Kosh Bilash Bagale, Ramesh Adhikari, Devaraj Acharya.

**Writing – original draft:** Kosh Bilash Bagale, Devaraj Acharya.

**Writing – review & editing:** Kosh Bilash Bagale, Ramesh Adhikari, Devaraj Acharya.

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
