## [Decision Letter · Decision Letter 0]

23 Sep 2022

Dear Mr Bagale,

Thank you very much for submitting your manuscript "Regional variation in knowledge and practice regarding common zoonoses in livestock farmers of selective districts in Nepal" for consideration at PLOS Neglected Tropical Diseases. As with all papers reviewed by the journal, your manuscript was reviewed by members of the editorial board and by several independent reviewers. In light of the reviews (below this email), we would like to invite the resubmission of a significantly-revised version that takes into account the reviewers' comments. 

We cannot make any decision about publication until we have seen the revised manuscript and your response to the reviewers' comments. Your revised manuscript is also likely to be sent to reviewers for further evaluation.

Sincerely,

Paul R. Torgerson

Academic Editor

Sergio Recuenco

Section Editor

Reviewer's Responses to Questions

**Key Review Criteria Required for Acceptance?**

**Methods**

-Are the objectives of the study clearly articulated with a clear testable hypothesis stated?

-Is the study design appropriate to address the stated objectives?

-Is the population clearly described and appropriate for the hypothesis being tested?

-Is the sample size sufficient to ensure adequate power to address the hypothesis being tested?

-Were correct statistical analysis used to support conclusions?

-Are there concerns about ethical or regulatory requirements being met?

Reviewer #1: The manuscript's main aim is to assess livestock farmers' knowledge and practices regarding a selection of zoonotic diseases. The paper sometimes does not flow and is not always very clear. If the reader is not familiar with the geography of Nepal, it is hard to understand what is meant by Mountains, Hills, and Terai; a map would be helpful. The methodological choices are not always explicitly stated (more details description of sample size, data collection, primary outcomes of the survey, and how they used the Delphi method – if possible, add a copy of the survey in the supporting document). Additionally, the study selected six diseases to include in the survey but did not state why they had chosen these as the primary outcome of the survey.

Reviewer #2: (No Response)

**Results**

-Does the analysis presented match the analysis plan?

-Are the results clearly and completely presented?

-Are the figures (Tables, Images) of sufficient quality for clarity?

Reviewer #1: In the discussion section, the author presented the defined the diseases; it would be clear for the reader (that could be not familiar) to have these definitions earlier in the text, with an explanation of why these diseases were selected (and maybe add a reference with the description). The discussion is rather long and repeats many of the results or issues raised in the results section. In the debate, the results are compared with previous studies, but sometimes it fails to stress what the comparison shows or stress – for the BTB, the comparison and the context are really good. It would be nice to have the same for the rest of the diseases if possible. In addition, it would be nice to summarize the ecological difference with a table for the primary outcomes (to stress better the take-home message).

Reviewer #2: (No Response)

**Conclusions**

-Are the conclusions supported by the data presented?

-Are the limitations of analysis clearly described?

-Do the authors discuss how these data can be helpful to advance our understanding of the topic under study?

-Is public health relevance addressed?

Reviewer #1: The conclusion is in line with the rest of the text. However, more stress on the limitation and prospects are not explicitly stated.

Reviewer #2: (No Response)

**Editorial and Data Presentation Modifications?**

Reviewer #1: The manuscript needs English revision in order to be suitable for publication.

Reviewer #2: (No Response)

**Summary and General Comments**

Reviewer #1: The aim of the manuscript is clear, and the topic is in line with the chosen journal. However, the manuscript lacks transparency and validity. More details on methodological choices should be included to solve these issues (sample size, data collection, use of the Delphi method). Additionally, the six diseases that were chosen lack an explanation, and the choice seems arbitrary. An answer is needed. 

A map would be a great addition due to the relevance of the geographical area for the manuscript's results. Some references are missing, such as the source for the disease definitions or the use of fertilizer, pesticides, or antibiotics (p. 8). 

 A summary of the main findings and key messages would reinforce the manuscript and clarify what the reader should retain. 

Carefully review that the acronyms are introduced before being used and then used constantly in the manuscript

Reviewer #2: (No Response)

PLOS authors have the option to publish the peer review history of their article (what does this mean?). If published, this will include your full peer review and any attached files.

Reviewer #1: No

Reviewer #2: Yes: Clovice Kankya
---

## [Decision Letter · Decision Letter 1]

12 Dec 2022

Dear Mr Bagale,

Thank you very much for submitting your manuscript "Regional variation in knowledge and practice regarding common zoonoses among livestock farmers of selective districts in Nepal" for consideration at PLOS Neglected Tropical Diseases. As with all papers reviewed by the journal, your manuscript was reviewed by members of the editorial board and by several independent reviewers. The reviewers appreciated the attention to an important topic. Based on the reviews, we are likely to accept this manuscript for publication, providing that you modify the manuscript according to the review recommendations. 

See reviewers comments. Some minor issues need to be corrected

Sincerely,

Paul R. Torgerson

Academic Editor

Sergio Recuenco

Section Editor

See reviewers comments. Some minor issues need to be corrected

Reviewer's Responses to Questions

**Key Review Criteria Required for Acceptance?**

**Methods**

-Are the objectives of the study clearly articulated with a clear testable hypothesis stated?

-Is the study design appropriate to address the stated objectives?

-Is the population clearly described and appropriate for the hypothesis being tested?

-Is the sample size sufficient to ensure adequate power to address the hypothesis being tested?

-Were correct statistical analysis used to support conclusions?

-Are there concerns about ethical or regulatory requirements being met?

Reviewer #1: (No Response)

**Results**

-Does the analysis presented match the analysis plan?

-Are the results clearly and completely presented?

-Are the figures (Tables, Images) of sufficient quality for clarity?

Reviewer #1: (No Response)

**Conclusions**

-Are the conclusions supported by the data presented?

-Are the limitations of analysis clearly described?

-Do the authors discuss how these data can be helpful to advance our understanding of the topic under study?

-Is public health relevance addressed?

Reviewer #1: (No Response)

**Editorial and Data Presentation Modifications?**

Reviewer #1: (No Response)

**Summary and General Comments**

Reviewer #1: The manuscript has been revised, and the comments have been considered. 

Nevertheless, some repetitions between the results and the discussion part slow the flow of the manuscript and result in a lengthy discussion. Therefore, I advise checking the information presented in the result and avoiding representing them in the debate. 

Some parts of the discussion would also benefit from references to support claims.

Lastly, there are some inconsistencies in how acronyms are introduced in the manuscript used. Therefore, it is advised to spell out the words before using the acronym and to be consistent with the same acronym (for example, Bovine TB vs BTV).

PLOS authors have the option to publish the peer review history of their article (what does this mean?). If published, this will include your full peer review and any attached files.

Reviewer #1: No

Figure Files:

Data Requirements:

Reproducibility:

References

---

## [Editor Report · Decision Letter 2]

9 Jan 2023

Dear Mr Bagale,

We are pleased to inform you that your manuscript 'Regional variation in knowledge and practice regarding common zoonoses among livestock farmers of selective districts in Nepal' has been provisionally accepted for publication in PLOS Neglected Tropical Diseases.

Best regards,

Paul R. Torgerson

Academic Editor

Sergio Recuenco

Section Editor

---

## [Editor Report · Acceptance letter]

30 Jan 2023

Dear Mr Bagale,

We are delighted to inform you that your manuscript, "Regional variation in knowledge and practice regarding common zoonoses among livestock farmers of selective districts in Nepal," has been formally accepted for publication in PLOS Neglected Tropical Diseases.

Best regards,

Shaden Kamhawi

co-Editor-in-Chief

Paul Brindley

co-Editor-in-Chief
